# Epigenetic Modifications of White Blood Cell DNA Caused by Transient Fetal Infection with Bovine Viral Diarrhea Virus

**DOI:** 10.3390/v16050721

**Published:** 2024-05-01

**Authors:** Hana Van Campen, Jeanette V. Bishop, Zella Brink, Terry E. Engle, Carolina L. Gonzalez-Berrios, Hanah M. Georges, Jessica N. Kincade, Dilyara A. Murtazina, Thomas R. Hansen

**Affiliations:** 1Department of Biomedical Sciences, Colorado State University, Fort Collins, CO 80523, USA; hana.van_campen@colostate.edu (H.V.C.); jeanette.bishop@colostate.edu (J.V.B.); zella.brink@colostate.edu (Z.B.); clgonz365@gmail.com (C.L.G.-B.); hanah.georges@yale.edu (H.M.G.); jkincade@colostate.edu (J.N.K.); dilyara.murtazina@colostate.edu (D.A.M.); 2Department of Animal Science, Colorado State University, Fort Collins, CO 80523, USA; terry.engle@colostate.edu; 3Currently at Department of Obstetrics, Gynecology and Reproductive Sciences, Yale School of Medicine, New Haven, CT 06510, USA

**Keywords:** bovine viral diarrhea virus, transient infection, immune system, epigenetics, pathway analysis, intrauterine growth restriction

## Abstract

Bovine viral diarrhea virus (BVDV) infections cause USD 1.5–2 billion in losses annually. Maternal BVDV after 150 days of gestation causes transient fetal infection (TI) in which the fetal immune response clears the virus. The impact of fetal TI BVDV infections on postnatal growth and white blood cell (WBC) methylome as an index of epigenetic modifications was examined by inoculating pregnant heifers with noncytopathic type 2 BVDV or media (sham-inoculated controls) on Day 175 of gestation to generate TI (*n* = 11) and control heifer calves (*n* = 12). Fetal infection in TI calves was confirmed by virus-neutralizing antibody titers at birth and control calves were seronegative. Both control and TI calves were negative for BVDV RNA in WBCs by RT-PCR. The mean weight of the TI calves was less than that of the controls (*p* < 0.05). DNA methyl seq analysis of WBC DNA demonstrated 2349 differentially methylated cytosines (*p* ≤ 0.05) including 1277 hypomethylated cytosines, 1072 hypermethylated cytosines, 84 differentially methylated regions based on CpGs in promoters, and 89 DMRs in islands of TI WBC DNA compared to controls. Fetal BVDV infection during late gestation resulted in epigenomic modifications predicted to affect fetal development and immune pathways, suggesting potential consequences for postnatal growth and health of TI cattle.

## 1. Introduction

Bovine viral diarrhea viruses (BVDVs) are enveloped, single-stranded, positive-sense RNA viruses classified as pestiviruses within the family Flaviviridae, with a worldwide distribution in domestic cattle and other ruminant species [1]. Transmission of BVDV occurs horizontally, resulting in acute infections, and vertically from the acutely infected pregnant cow to her fetus [2]. The outcome of acute postnatal BVDV infections varies from clinically inapparent to severe hemorrhagic disease and peracute death depending on the virus strain [3,4]. Vertical transmission and fetal BVDV infection result in embryonic losses, a range of congenital malformations, abortions, stillbirths, and the birth of weak or non-viable calves [5]. Fetal BVDV infection between Days 30 and 125 of gestation, prior to the full development of the fetal immune system, results in the birth of immunotolerant, persistently infected (PI) calves [6,7]. PI cattle have reduced viability and longevity [8]; however, PI cattle are the main source of horizontal BVDV infections and maintenance of the virus in cattle populations by virtue of continual virus shedding during their lifetime.

In contrast, BVDV infections after 150 days of gestation result in a transient fetal infection (TI) when the more fully developed immune system can respond to and clear the virus. Previously, we have shown that when pregnant heifers are inoculated with a noncytopathic (ncp) type 2 BVDV on Day 175 of gestation, the fetuses mount a vigorous innate immune response, and genes bridging the innate and adaptive responses are activated [9]. The B cell component of the adaptive immune response in TI fetuses is also fully functional as inferred from the presence of BVDV-specific neutralizing antibodies present in serum collected at birth prior to the ingestion of colostrum. While the health, immune system, and mechanisms of immunotolerance of PI calves have been studied in detail, less is known about the impact of fetal TI BVDV infections.

In epidemiologic studies, Munoz-Zanzi et al. (2003) obtained blood samples from dairy calves prior to the ingestion of colostrum and identified congenitally infected (CI; synonymous with TI) calves by the presence of BVDV SN antibodies (titers ≥ 4) after birth [10]. In this study, CI calves were 2.3 times more likely to suffer an illness requiring treatment compared to non-CI calves. Waldner and Kennedy (2008) used BVDV serum neutralizing (SN) antibody titers at weaning to distinguish potential late-term BVDV-infected calves and assessed differences in weights with seronegative calves from the same herd [11]. They reported that age-adjusted calves with type 1 BVDV SN titers ≥ 1000 were 13 kg lighter in weight than calves with BVDV type 1 SN titers < 1000, and calves with type 2 BVDV SN titers > 1000 were 15.1 kg lighter than calves with BVDV type 2 SN titers < 1000. The difference in weaning weights between calves with high BVDV antibody titers was striking; however, the presence of high anti-BVDV SN titers at weaning does not definitively distinguish between TI calves and calves infected with BVDV between birth and weaning. The authors speculated that the weaning weight differences would be an additional economic loss in BVDV-infected beef herds. Although these epidemiologic studies showed that TI calves had a greater prevalence of disease and were lighter in weight than uninfected age-adjusted calves in their cohort, the suspected specific effects of fetal infection on the growth, immune system, postnatal health, and feedlot performance have not been confirmed experimentally.

Epigenetic changes found previously in the DNA of PI fetal calf spleens were associated with genes and pathways relevant to the immunological and congenital defects described for PI calves [12]. Therefore, we hypothesized that late-term BVDV infection of the bovine fetus might also cause epigenetic modifications in TI calves predictive of decreased growth and impaired immune systems. Furthermore, we posit that DNA in white blood cells (WBC) during fetal life may reflect more general epigenetic modifications and phenotypes not only in immune cells but also in other cells and organ systems of the body. To this end, TI and control calves were generated by inoculating pregnant heifers with a noncytopathic (ncp) type 2 BVDV suspended in media or media alone on Day 175 of gestation. The birth weights and methylomes of TI and control calves were compared.

## 2. Materials and Methods

### 2.1. Animals

Unvaccinated, yearling Hereford heifers were confirmed to be seronegative for antibodies to BVDV1 and BVDV2 by serum neutralization (SN) tests and negative for BVDV antigen in ear notch extracts by BVDV ELISA (IDEXX Laboratories, Westbrook, ME, USA). Heifers were housed at the Agriculture, Research, Development, and Education Center—Colorado State University (ARDEC-CSU). Each heifer was vaccinated twice with Clostridium Chauvoei-Septicum-Haemolyticum-Novyi-Sordelli-Perfringins Types C and D (Ultrabac, Zoetis, Kalamazoo, MI, USA) one month apart. The heifers were moved to the Animal Reproduction & Biotechnology Laboratory (ARBL) cattle facilities at approximately 13.5 months of age. Beginning at 14 months of age, heifers were estrus synchronized with intravaginal progesterone inserts (EAZI-BREED, Zoetis) for 14 days, followed by PGF2α (Lutalyse con, Zoetis) i.m. and the application of a heat-detection patch (Estrotect, Spring Valley WI, USA). Heifers with activated heat-detection patches (<50% of the patch activated; 60 to 72 h after PGF2α) were artificially inseminated (AI) with female sexed semen from a single Angus bull (Select Sires MidAmerica, Logan, UT, USA). Pregnancy was diagnosed by trans-rectal ultrasonography at 32 to 38 days and confirmed at 60 to 65 days post-AI at which time fetal sex was determined. Heifers were vaccinated 6 and 3 weeks prior to their predicted calving date with coronavirus-rotavirus vaccine Clostridium perfringens types C–E. coli bacterin toxoid (ScourGuard 4KC, Zoetis). All experiments were approved by the Institutional Animal Care and Use Committee at Colorado State University (Protocol approval number: 1656, 27 April 2021).

### 2.2. Virus and Inoculum

The BVDV virus 96B2222, a ncp BVDV2 [13] stock, was grown in bovine turbinate cells (BT) in Dulbecco’s Modified Eagle Medium (DMEM) + 4.5 g/L D-glucose + l-glutamine (Gibco/Thermo Fisher, Waltham, MA USA, Cat# 11965-092) + 2% horse serum (HS) (Sigma-Aldrich, St Louis MO, USA, Cat# H1138) and an inoculum of 4.0 log_10_TCID_50_/_mL_ of ncp BVDV2 96B2222 was prepared in DMEM + 2% horse serum (HS) [14].

### 2.3. Experimental Design and Blood Collection

Heifers were determined to be pregnant by ultrasound 60 to 65 days post-AI. Fetal viability was confirmed by rectal palpation on Days 150 to 165 post-AI. Heifers with similar estimated calving dates were randomly placed into one of two groups, controls or BVDV-infected. Controls were generated by being inoculated intranasally on Day 175 of gestation with 4 mL of DMEM+ 2% HS (media only), to generate control heifer calves (*n* = 12) while infected heifers were inoculated with 4 mL of 4.0 log_10_TCID_50_ of ncp BVDV2 strain 96B2222 in DMEM + 2% HS to produce TI heifer calves (*n* = 11) as previously described [14]. Heifers were observed for signs of parturition and the newborn calves were ear-tagged, weighed, and jugular vein blood samples were collected in vacutainer tubes containing K_2_EDTA or no anticoagulant (Becton, Dickinson and Company, Franklin Lakes, NJ, USA) prior to the calves standing and nursing. A second jugular vein blood sample was collected from each calf at 24 to 60 h of age. Blood tubes were placed on ice and processed within 1 h of collection. Processing consisted of centrifugation at 1200× *g* for 10 min at 4 °C. Serum was separated for immunoglobulin concentration determination.

### 2.4. WBC Preparation for RNA and DNA Extraction

Whole blood was collected in either blood collection tubes for serology or containing K_2_EDTA (Becton, Dickinson and Company, Franklin Lakes, NJ 07417). Blood tubes were centrifuged at 377× *g* for 10 min at 4 °C (Eppendorf 5804R, Enfield, CT, USA). The separated serum was placed into microcentrifuge tubes and stored at −20 °C for serology. Tubes containing K_2_EDTA were centrifuged at 300× *g* for 10 min at 4 °C. For each sample, the buffy coat was transferred into tubes containing 5 mL of an ammonium–chloride–potassium lysing buffer (ACK) (KD Medical, Columbia, MD, USA, Cat# RGF-3015) and incubated at room temperature for 5 min. The WBCs were pelleted by centrifugation at 42× *g* for 10 min at 4 °C, resuspended in 2.5 mL of ACK, and centrifuged at 300× *g* for 10 min at 4 °C. Following centrifugation in ACK, the supernatant was decanted, and the remaining WBC pellet was washed by resuspension in 1X PBS, (pH 7.4) followed by centrifugation at 300× *g* for 10 min at 4 °C. Following centrifugation, the supernatant was decanted, and the WBC pellets were resuspended in 1 mL of PBS for DNA extraction or 1 mL of Trizol (Ambion Life Technologies, Carlsbad, CA, USA) for RNA extraction.

### 2.5. Serology

Serum-neutralizing (SN) antibody titers were determined in a microtiter plate format using cytopathic (cp) BVDV2 (296c) and BTs as indicator cells [15]. Each serum dilution was tested in duplicate and 100 TCID_50_/25 µL of the test virus was added to each well. The plates were incubated for 1 h at 37 °C, 5% CO_2_ before adding 1 × 10^4^ BT/well. The BTs in each well were scored for cytopathic effect after an additional 72 h of incubation. Each titration assay included duplicates of the TI and control calf sample, a positive control serum of known titers, cell controls, and the inoculum. The SN titer was the highest dilution of serum that prevented cytopathology.

### 2.6. IgG Quantification RID

Serum samples collected from calves 24 to 60 h after birth were assayed for IgG concentrations by radial immuno-diffusion (RID) assay (Colorado State University Veterinary Diagnostic Laboratory, USA) to confirm adequate colostrum absorption.

### 2.7. RNA Extraction and BVDV RT-PCR

RNA was extracted from WBCs using TRIzol reagent (Ambion, Carlsbad, CA, USA), treated with 6.8 Units of RNase-Free DNase I (Qiagen, Germantown, MD, USA, Cat# 79254) per sample and purified using RNeasy MinElute Cleanup Kit columns (Qiagen, Cat# 74204) according to the manufacturer’s instructions. The quantity and quality of the RNA were assessed using a Nanodrop ND-1000 spectrophotometer (Thermo Scientific, Waltham, MA, USA) and all samples had 260/280 ratios > 2.03. To confirm that the calves were not infected with BVDV, RNA extracted from WBCs in blood samples collected 1 week after birth were assayed for the presence of BVDV RNA by RT-PCR. RNA extracted from the ncp BVDV2 (96B2222) used to inoculate the pregnant heifers also was used as a positive control for BVDV RT PCR amplification. Briefly, cDNA was produced from 1 ug of RNA using Bio-Rad (Boulder, CO, USA) iScript Reverse Transcription Supermix (Cat# 1708841) at 46 °C for 1 h followed by 94 °C for 4 min (Mastercycler, Eppendorf, Hamburg, Germany). The primers used to detect BVDV RNA were as follows: forward, 5′-CAT GCC CAT AGT AGG AC-3′; reverse, 5′-CCA TGT GCC ATG TAC AG-3′. These amplify all BVDV1 and BVDV2 isolates [16]. The PCR reaction for each sample consisted of 1.5 µL of cDNA, 0.5 µL of each of the forward and reverse primers (3 µM), and 22.5 µL of Invitrogen PCR Supermix (Thermo Fisher Scientific, Waltham, MA, USA). The PCR reaction consisted of 41 cycles of 94 °C for 10 s, 50 °C × 15 s, 72 °C × 30 s for 41 cycles, followed by 1 cycle of 72 °C for 10 min and then held at 4 °C. Products were separated on a 2% agarose gel containing Gel Red (Biotium. Fremont, CA, USA, Cat# 41003) and visualized with a Molecular Imager ChemDoc XRS+ with Image Lab software (Version 6, BioRad, Hercules, CA, USA).

### 2.8. Reduced Representation Bisulfite Sequencing

DNA was extracted from WBCs processed from whole blood (see Section 2.4 DNA extraction) using a Qiagen DNeasy Blood and Tissue Kit as previously described [12]. Genome-wide classic reduced representation bisulfite sequencing (RRBS/methyl-seq) was performed for 5 control and 5 TI randomly selected WBC DNA samples by the methods provided by Zymo Research (Zymo Research, Irvine, CA, USA). Briefly, DNA samples were digested with 30 units of MspI (NEB; Ipswich, MA, USA), the fragments were then purified with DNA Clean & Concentrator-5 (Zymo Research; Irvine, CA, USA) and ligated to adapters with the replacement of cytosine with 50-methyl-cytosine according to Illumina’s guidelines. Fragments greater than 50 base pairs were recovered and treated with bisulfite using the EZ DNA Methylation-Lightning Kit (Zymo Research; Irvine, CA, USA). Samples were subjected to PCR with Illumina indices. The size and concentrations of purified products were confirmed with the Agilent 2200 TapeStation prior to sequencing on an Illumina platform.

### 2.9. Methylation Bioinformatics and Pathway Analysis

DNA samples from randomly selected TI (*n* = 5) and control (*n* = 5) calves were analyzed by Zymo Research (Irvine, CA, USA). Raw BAM files received from Zymo Research were analyzed in R (version 4.2.0) [17] using the methylKit (version 1.24.0) [18]. Differential DNA methylation was calculated by comparing the proportion of methylated cytosines in a test sample relative to control DNA. The bases or regions with different methylation proportions across samples were identified as differentially methylated cytosines (DMCs) and differentially methylated regions (DMRs), respectively. DMCs were considered significant with *p* < 0.01, and a 25% or greater difference in methylation and DMRs was considered different with *p* < 0.01 and a 15% difference in methylation between TI and control samples. Logistic regression was used to model methylation levels in relation to the sample groups and the variation between replicates. Gene IDs were identified using the genomation (version 1.30.0) [19] R package and *Bos taurus* reference genome ARS-UCD1.3. Quality control plots and gene ontology plots were generated with clusterProfiler (v4.6.2), pathview (version 1.38.0), and gage (version 2.48.0) R packages. Raw fasta files are available in the NCBI GEO Database (accession number GSE255721). Pathway analysis for hyper- and hypomethylated DMCs and DMRs were analyzed using Ingenuity Pathway Analysis (IPA; QIAGEN Inc., https://www.qiagenbioinformatics.com/products/ingenuity-pathway-analysis, accessed on 1 March 2024, Germantown, MD, USA) as previously described [12] and the Kyoto Encyclopedia of Genes and Genomes (KEGG; https://www.genome.jp/kegg/, accessed on 1 March 2024) [20].

### 2.10. Statistical Analyses

The numbers of control heifer calves (*n* = 12) and TI heifer calves (*n* = 11) allow differentiation of proportions of 0.9 vs. 0.1 at a power of 80% (Fisher’s exact test). Statistical analyses were performed in GraphPad Prism 9 (GraphPad Software, San Diego, CA, USA). The data were checked for normality using D’Agostino and Pearson, Anderson–Darling, Shapiro–Wilk, and Kolmogorov–Smirnov tests. Unpaired, two-tailed Student’s t-tests were used to compare calf weights and gestational lengths for control and TI calves. Differences between control and TI calves were considered significant when *p* < 0.05. Data are presented as the mean ± standard error of the mean (SEM).

## 3. Results

### 3.1. Heifer Inoculations, TI, and Control Calf Weights and Serology

All BVDV-inoculated heifers seroconverted by Day 14 post-inoculation and all control heifers remained seronegative. Control and BVDV-inoculated heifers did not exhibit clinical signs of disease. A total of 12 control and 11 TI calves stood and nursed within 3 h of birth. The mean weight of newborn TI calves (mean ± SEM; 27.8 ± 0.96 kg) was less than the mean weight of control calves (31.85 ± 1.18 kg) (*p* < 0.05) (Figure 1). There was no difference in the gestational length of control (mean ± SEM; 274.0 ± 1.15 days) and TI calves (274.6 ± 1.45 days) (*p* = 0.58). All control calves were seronegative, and all TI calves were seropositive for type 2 BVDV SN antibodies (≥128) at birth. WBC from all calves were negative for BVDV RNA by RT-PCR (Appendix A). All calves had serum IgG immunoglobulin levels of >2200 mg/dL 24 to 60 h after birth, which was evidence of adequate passive transfer of colostrum antibodies.

### 3.2. Reduced Representation Bisulfite Sequencing: Overview of DMCs and DMRs

Classic RRBS of WBC DNA indicated no differences in whole genome global methylation levels between TI and control calves. Principal components analysis (PCA) did not demonstrate a clear separation of samples by treatment which supports the lack of global percent methylation differences between control and TI samples. There were 2349 DMCs including 1277 hypomethylated and 1072 hypermethylated cytosines in TI compared to control calves. The DMCs are represented in the heatmap (Figure 2).

Of the DMCs, 15% were in exons, 17% in promoters, 30% in intergenic regions, and 37% in introns. DMCs were found in all chromosomes (Figure 3). A total of 173 DMRs was also identified between TI and control calves with 84 DMRs located in promoters and 89 DMRs in islands. It is assumed in these studies that hypermethylation potentially results in the downregulation of affected genes, whereas hypomethylation results in the upregulation of affected genes.

### 3.3. Pathway Analysis of DMCs and DMRs

The methylome data include DMCs and DMRs that potentially affect gene expression in multiple pathways in the WBC DNA of TI calves compared to controls. An overall view of the top 20 major canonical pathways and genes affected by hyper- and hypomethylation is presented in Figure 4. A more detailed analysis of methylation was conducted by examining both hyper- and hypomethylated genes and associated pathways in specific upstream regulators and disease, and biological function pathways (Figure 5; also see Appendix A for a complete list of canonical pathways). Figure 6 provides IPA legends, network shapes and path designer shapes for the selected canonical pathways highlighted herein.

Of the top twenty canonical pathways, several pathways may be grouped together based on their function or the organ systems affected, such as (1) nervous system development, GABA receptor signaling, Axonal Guidance Signaling, Opioid signaling, and Ephrin receptor signaling pathways; (2) immune system development, WNT/β-catenin signaling, G-Protein Coupled Receptor Signaling, Role of Macrophages, Fibroblasts and Endothelial Cells in Rheumatoid Arthritis, and Th1 (Figure 7) and Th2 (Figure 8) activation pathways; and (3) embryonic differentiation and development, Transcriptional Regulatory Network in Embryonic Stem Cells (Figure 9), Rho family GTPases, and Human Embryonic Stem Cell Pluripotency (Figure 10).

Additional pathways affecting the development and activation of the immune system were impacted by TI fetal BVDV infection. These pathways included the fMLP Signaling in Neutrophils; CCR3 Signaling in Eosinophils; and IL-4, IL-8, IL-13, IL-15, IL-22, and IL-20 signaling. Pathways impacting cardiac development included Cardiac Hypertrophy Signaling (Appendix A), Cardiac β-adrenergic Signaling, Factors Promoting Cardiogenesis in Vertebrates, Cardiac Hypertrophy Signaling, the Role of NFAT in Cardiac Hypertrophy, and Dilated Cardiomyopathy Signaling. Pathways associated with bone development and disease included the Role of Osteoblasts in Rheumatoid Arthritis Signaling; the Role of Osteoblasts, Osteoclasts, and Chondrocytes in RA (Appendix A); Chondroitin Sulfate Biosynthesis; and RANK Signaling in Osteoclasts. Pathways affecting the liver included Hepatic Fibrosis Signaling and Hepatic Stellate Cell Activation. The Pulmonary Healing Signaling and Pulmonary Fibrosis Idiopathic Signaling pathways were ascribable to the lung.

Genes containing hypermethylated CpGs and predicted to be downregulated in TI calves relative to controls impacted multiple pathways. These genes included but were not limited to (1) the secreted growth factors *WNT3A*, *WNT7A*, *WNT7B*, and *WNT10A* in the Transcriptional Regulatory Network in Embryonic Stem Cells; Cardiac Hypertrophy; Pulmonary Healing; the Role of Osteoblasts, Osteoclasts and Chondrocytes in Rheumatoid Arthritis; the Role of Macrophages; Fibroblasts and Endothelial Cells in Rheumatic Arthritis pathways (Figure 9, Figure 11, Appendix A); (2) the multifunctional cytokine leukemia inhibitor factor (*LIF)* in the Transcriptional Regulatory Network in Embryonic Stem Cells and Cardiac Hypertrophy pathways (Figure 9 and Appendix A); (3) members of the fibroblast growth factor family, *FGF3*, *FGF19*, and receptor *FGFR1*; the transforming growth factor beta (*TGFB1*) in the Transcriptional Regulatory Network in Embryonic Stem Cells; Human Embryonic Stem Cell Pluripotency; Cardiac Hypertrophy pathways (Figure 9, Figure 10, and Appendix A); the calcium-binding proteins 1 and 4 (*CALM1/4*) in the Cardiac Hypertrophy; the Role of Macrophages, Fibroblasts and Endothelial Cells in Rheumatoid Arthritis; fMLP Signaling in Neutrophils; CCR3 Signaling in Eosinophils; and the Role of Osteoblasts, Osteoclasts and Chondrocytes in Rheumatoid Arthritis pathways (Figure 11 and Appendix A); and (4) the transcription factors; the nuclear factor of activated T cells 1 (*NFATc1*) and 4 (*NFATc4*) in Th1 and Th2 Activation; Cardiac Hypertrophy; the Role of Osteoblasts, Osteoclasts and Chondrocytes in RA; and fMLP Signaling in Neutrophils pathways (Figure 7, Figure 8, Appendix A).

Genes containing hypomethylated CpGs and predicted to be upregulated in TIs compared to controls that appeared in multiple pathways included (1) the secreted growth factors *WNT4*, *WNT5B*, *WNT7A*, and *WNT10A*; (2) the WNT receptor, Frizzled (*FZD1*), in the Transcriptional Regulatory Network in Embryonic Stem Cells, Human Embryonic Stem Cell Pluripotency, Factors Promoting Cardiogenesis in Vertebrates (Figure 9 and Figure 10); (3) members of the MAP kinase family, *ERK1/2/MAPK3*, which phosphorylate nuclear targets in Cardiac Hypertrophy, fMLP Signaling in Neutrophils, CCR3 Signaling in Eosinophils, Growth Hormone Signaling, and multiple other pathways (Appendix A); (4) protein kinases *PRKCB*, *PRKX*, and ITPKB in the Human Embryonic Stem Cell, Role of Osteoclasts in Rheumatoid Arthritis, Cardiac Hypertrophy pathways (Figure 10, Appendix A); (5) transcription factors Forkhead Box Protein D3 (*FOXD3*) in the Transcription Regulatory Network in Embryonic Stem Cells (Figure 9) and cAMP response element-binding protein (*CREB*) in Factors Promoting Cardiogenesis in Vertebrates pathways; and (6) members of the rat sarcoma virus (RAS) oncogene family of GTPases *RAB4A*, *RAB26*, *RAB40C*, and *RABGAP1L*.DMRs located in gene promoters may also influence the transcription of these genes (Figure 12).

Eighty-four DMRs were found in gene promoters of TIs versus controls (Figure 12). Gene promoters containing hypermethylated CpGs were found in *CALM4*, *FGR*, and *BLK*. Gene promoters with hypomethylated CpGs were found in matrix metalloprotein 9 (*MMP9*), interleukin 11 receptor subunit A (*IL11RA*), the zeta chain of T cell receptor-associated protein kinase 70 (*ZAP70*), and lymphocyte expansion molecule (*LEXM*) genes. The canonical pathways for these DMRs analyzed in IPA include but are not limited to the immune response-related pathways, the Neutrophil Extracellular Trap Signaling, IL-15 Production, IL-15 Signaling, IL-13 Signaling, Activating JAK and STAT6, IL-6 Family Signaling, and Neutrophil Degranulation pathways. An additional 89 DMR islands were identified within genes in the TI methylome. The CpG islands containing hypermethylated CpGs included Fc gamma receptor1a (*FCGR1A*), *ZAP70*, and interleukin 9 receptor (*IL9R*). Genes containing CpG islands with hypomethylated CpGs included phosphoinositide-3-kinase interacting protein 1 (*PIK3IP1*), an RAS oncogene family member (*RAB20*), and *LEXM*. Canonical pathways predicted to be affected by IPA based on these DMRs included Interleukin-9 Signaling, IL-1 Signaling, IL-15 Production pathways related to the immune response, and Cardiac β-adrenergic Signaling and Cardiac Hypertrophy Signaling pathways referable to cardiac development.

### 3.4. Pathway Analysis of DMCs and DMRs Using KEGG

The DMC and DMR data from TI WBC DNA were analyzed using KEGG in the context of the bovine genome (ARS-UCD1.2/BosTau9) [20]. The hyper- and hypomethylated CpGs in TI WBC DNA relative to controls were found in common with multiple pathways identified in IPA such as the WNT Signaling, ERK/MAPK Signaling, and Calcium Signaling pathways which control basic cellular processes involved in cellular growth, differentiation, and embryonic development. KEGG analysis identified multiple pathways involving the immune system including B Cell Receptor Signaling (Figure 13). Other pathways identified by KEGG analysis included T Cell Receptor Signaling, Cytokine–Cytokine Receptor Interaction (Appendix A), Chemokine Signaling, Hematopoietic Cell Lineage, T Cell Receptor Lineage, and TNF signaling pathways that were similar to pathways found in the IPA analyses. Pathways involved in growth and development included but were not limited to the Metabolic, Biosynthesis of Fatty Acids, Fatty Acid Metabolism, Mineral Absorption, and Glutathione Metabolism pathways.

Also similar to the IPA analysis, KEGG found DMRs in the promoters of 20 genes including Src family tyrosine kinases *BLK* in the B Cell Receptor Signaling pathway (Figure 13), *FGR* in the Chemokine Signaling pathway, interleukin 11 receptor subunit alpha (*IL11RA*) in the Cytokine–Cytokine Receptor Interaction pathway (Appendix A), Interleukin 2 receptor subunit beta (*LOC510185*) in the Th1 and Th2 Receptor Interaction pathway, and Matrix Metallopeptidase 9 (*MMP9*) in the TNF Signaling pathways. Of these genes, the DMRs in *BLK* and *FGR* contain hypermethylated CpGs. KEGG analysis also found DMRs in the islands within 19 genes including *IL9R* in the Cytokine–Cytokine Receptor Interaction, JAK-STAT Signaling, and the Hematopoietic Cell Lineage pathways; *ZAP70* in the Natural Killer Cell-Mediated Cytotoxicity and the T Cell Receptor Signaling pathways; Mitogen-Activated Protein Kinase 6 (*MAP2K6*) in the GnRH Signaling; the Growth Hormone Synthesis, Secretion, and Action pathways; and Calcium Voltage Channel (*CACNG1*) in the Hypertrophic, Arrhythmogenic Right Ventricular, and Dilated Cardiomyopathy pathways. The DMRs in islands of *IL9R* and *ZAP70* contained hypermethylated CpGs and the DMRs of *MAP2K6*, and *CACNG* contained hypomethylated CpGs.

## 4. Discussion

### 4.1. Decreased Birth Weight and Evidence for an Active Immune Response in TI Calves at Birth

Transient fetal infection with BVDV induced by maternal inoculation with an ncp type 2 BVDV on Day 175 of gestation reduced the mean birth weight of TI calves compared to controls. These findings agree with previous epidemiologic observations in beef calves with BVDV SN antibody titers ≥ 256 at weaning that weighed 8.6 to 15. kg less than calves with lower SN titers [11]. The decreased birth weight in TI calves reported here is likely a continuation of growth restriction that began in utero. At birth, the TI calves were seropositive for BVDV-specific SN antibodies and were negative for BVDV RNA, confirming that they had been infected with BVDV in utero, mounted an immune response, and had cleared the virus infection as previously described [21]. These findings are evidence of a functional and effective immune system in the TI fetuses on Day 175 of gestation.

### 4.2. Transient Fetal BVDV Infection Causes Epigenetic Modifications in WBC DNA

The methylomes of WBCs from five TI calves were compared to those of five controls to identify hyper- and hypomethylated DMCs, DMRs, genes, and pathways potentially involved in the decreased birth weights and increased susceptibility to disease associated with TI BVDV infections in previous studies. The total number of DMCs (2349) in the TI WBC DNA is comparable to the 2641 DMCs found in the DNA of spleens from PI fetuses on Day 245 of gestation [12], indicating that epigenetic changes also occur in older bovine fetuses in response to BVDV infection. However, the number of hypomethylated DMCs in WBC DNA compared to controls was approximately 1.8 times greater in TIs compared to that of PI fetal spleens (1277 vs. 691) and the number of hypermethylated DMCs commensurately less in TIs compared to the splenic DNA of PI fetuses (1072 vs. 1951) [12]. The increased hypomethylated and decreased hypermethylated CpGs in TI calves compared to PI fetuses may reflect some resistance to or impairment of DNA methylation processes in the older (TI) fetuses at the time of fetal BVDV infection. It is possible that the active immune response of these older fetuses inhibited the hypermethylation of their WBC DNA relative to the controls. The mechanism by which the immune response to BVDV might interfere with DNA methylation is currently unknown.

### 4.3. Pathway Analysis of TI WBC Methylome

Analysis of the DMC data in both IPA and KEGG yielded links between the affected genes and clinical observations of impaired immune system development and growth previously described for TI calves [10,11]. Several pathways suggest the dysregulation of immune cell functions in TI calves. In the Th1 and Th2 Activation pathways (Figure 8 and Figure 9), multiple cytokines produced by Th2 cells were predicted to be downregulated including IL-3, IL-5, IL-9, IL-10, IL-13, and IL-31. These cytokines have a wide range of actions including promoting the growth, differentiation, and activation of T cells (IL-3, IL-9, IL-31), eosinophils (IL-3, IL-5, IL-13), and B cells (IL-5) [22,23,24,25]. IL-10 has regulatory functions and was predicted to be downregulated in both Th1 and Th2 cells. Produced in response to LPS, IFNγ, and other mediators, IL-10 inhibits the release of cytokines, antigen presentation, and phagocytosis by monocytes/macrophages, thus dampening the inflammatory response [26]. Inhibition of IL-10 production in TI animals would result in the loss of control over the immune response triggered by microbial infections leading to chronic inflammation, collateral damage to tissues, aberrant metabolism, and severe disease [27,28]. Other cytokines including interleukin-11 (IL-11), lymphotoxin-α (LTA), tumor necrosis family member 11 (TNSF11), chemokine receptor 3 (CXCR3), and C-C motif chemokine receptor 5 (CCR5) are predicted to be upregulated in Th1 cells. These cytokines have a broad range of actions including controlling lymphoid organ development (LTA), differentiation and activation of osteoclasts and T cells (TNFSF11), T cell memory (CXCR3), and the coordination of immune responses (CCR5) [29,30,31]. The gene for the co-receptor of T cell receptors, *CD8A*, also hypomethylated, is an important mediator of multiple immune cell interactions [32]. Upregulation of *CD8A* may reflect the activation of CTLs responsible for killing and clearing BVDV-infected cells during fetal infection. The hypomethylation and predicted upregulation of these genes may be a residual manifestation of the active immune response of TI fetuses previously reported to occur 15 days post-maternal inoculation with BVDV on Day 175 of gestation [14]. The predicted activation of immune cell production and differentiation in the methylome of TI calves contrasts with the inactivation of the innate and adaptive immune responses demonstrated in PI fetuses [33]. In summary, the DMGs in immune-related pathways suggest that BVDV infection at this later stage of gestation caused key genes in the immune system to be epigenetically modified, potentially dysregulating their expression, and causing impaired responses to microbial infections during post-natal life.

### 4.4. Key DMGs in TI WBC DNA

The key genes, *WNT3A*, *WNT7A/7B*, *WNT10A*, *LIF*, and *NFATc1*/4 contained hypermethylated DMCs in both TI WBC and PI splenic DNA. Members of the WNT family of glycoproteins exert their effects by binding to Frizzled receptors leading to activation of T cell factor (TCF) which regulates the expression of target genes [34]. Downregulation of WNT activity was predicted by IPA to inhibit embryonic stem cell pluripotency; mesoderm, endoderm, ectoderm, and extraembryonic tissue differentiation and development; and neurogenesis, extracellular matrix accumulation, and osteoclast function. Leukemia inhibitor factor (LIF) has a broad range of biological functions involving the neuronal, hepatic, endocrine, inflammatory, and immune systems and also regulates embryonic implantation and ESC pluripotency [35]. LIF binds to its receptor (LIFR) activating the JAK/STAT3, PI3K/AKT, ERK1/2, and mTOR signaling pathways. Inhibition of *LIF* was predicted to downregulate ID1-mediated functions including angiogenesis and the growth of a broad array of tumor cells. Collectively, hypermethylation and potential downregulation of *WNTs* and *LIF* were predicted to decrease fetal growth and differentiation. CALM1 and CALM4 are Ca^2+^ sensing proteins found in several major signaling pathways including the RAS, calcium, cAMP, adrenergic signaling in cardiomyocytes, and phosphotidylinositol signaling pathways. The latter functions in the contraction and relaxation of skeletal, smooth, and cardiac muscles. Downregulation of *CALM1* is predicted to cause electrical dysfunctions in muscle and is associated with cardiac arrhythmias; however, redundancy exists in the form of CALM2 and CALM3, which may mitigate this effect [36]. CALM4 is also a component of the intermicrovillar adhesion complex associated with the brush border cells of intestinal microvilli [37]. Inhibition of CALM4 perturbs brush border formation leading to malformation of intermicrovillar adhesion complexes dysregulating intestinal epithelial cell formation and function with potential consequences for the absorption of nutrients. NFATc1 and NFATc4 are components of the DNA-binding transcription complex in activated T cells and play key roles in immune responses including IL-2 and IL-4 cytokine induction. NFATc1 is also involved in cardiac valve formation and is required for osteoblastogenesis, osteoclastogenesis, and differentiation. Downregulation of *NFATc1* and NFATc4 was predicted to interfere with T cell activation and cardiac and bone development.

The following genes were hypomethylated in TIs compared to controls. Phosphoinositide-3-kinase (P3K) is a subunit of phosphoinositide 3 kinases and functions in the activation of AKT1 in multiple intracellular pathways affecting cell growth, proliferation, differentiation, migration, and secretory functions. Upregulation of *PI3K* is predicted to inhibit cell growth and differentiation in these pathways (Figure 7, Figure 8, Figure 9, Figure 10 and Appendix A). SMAD family member 7 (SMAD7) is a negative regulator of TGF-β signaling affecting multiple pathways with consequences for embryonic development, the inflammatory and immune response, and metabolism. Upregulation of *SMAD7* is predicted to have inhibitory effects on the transcription factors NANOG, SOX2, and POU5F4 with negative effects on embryogenesis including mesoderm, endoderm, and ectoderm differentiation and development (Figure 10) [38]. Upregulation of transcription factor CREB is predicted to be involved in cardiogenesis through the activation of *TGFB2.* Upregulation of transcription factor *FOXD3* is predicted to inhibit mesoderm, endoderm, ectoderm development, and differentiation through the activation of *POU5F1*, *SOX2*, and *NANOG* (Figure 9 and Figure 10). The RAS oncogene family of GTPases *RAB4A*, *RAB26*, *RAB40C*, and *RABGAP1L* are hypomethylated and potentially upregulated in TI animals. Located in the Golgi and ER, these GTPases regulate intracellular vesicle trafficking including exocytosis, endocytosis, and intracellular protein transport processes that are important in protein and lipid metabolism. RAB20, located in the Golgi apparatus and phagocytic vesicles, is involved in the fusion of lysosomes with phagosomes and phagosome acidification. Upregulation of *RAB20* may impact normal cellular processes such as phagocytosis and degradation of pathogens by macrophages. Flaviviruses including BVDV enter cells through endocytosis and uncoating within acidified endosomes [39]. Increased RAB20 activity along with other hypomethylated RAB family members may enhance the entry and uncoating of BVDV in cells such as macrophages for which these viruses have a tropism.

KEGG analysis of DMRs in promoters identified the hypermethylated gene *MMP9* in multiple pathways. MMP9 a metallopeptidase produced by macrophages, degrades the extracellular matrix and activates cytokines and chemokines involved in leukocyte migration, neutrophil functions, tissue remodeling, embryogenesis, bone development, and angiogenesis [40]. Inhibition of *MMP9* is predicted to impact the immune response and bone and heart development. The DMRs in islands within *IL9R* and *ZAP70* contained hypermethylated CpGs, suggesting the potential for downregulation of cytokine signaling and inhibition of T and NK cell functions. The DMRs of *MAP2K6* and *CACNG* contain hypomethylated CpGs, indicating that these genes are potentially upregulated. Upregulation of *MAP2K6* may affect growth through its effects on GnRH and growth hormone expression. Upregulation of *CACNG1* would potentially affect RAB20 functions, as discussed above.

### 4.5. Comparison of TI WBC and PI Spleen Methylomes

Hyper- and hypomethylated DMCs were found in genes and pathways related to the development of the nervous system, immune system, heart, liver, lung, and bone in the methylome of TI WBCs. In a previous study of the methylome of PI fetal spleens obtained on Day 245 of gestation, IPA also predicted pathways for diseases affecting the cardiovascular, neurologic, respiratory, skeletal muscle, hepatic, and reproductive systems [12]. Congenital defects in these organs have been described in PI cattle but have not been described in TI cattle. At present, the identification of TI cattle is problematic as infection with BVDV during late gestation can only be determined by identifying BVDV-specific antibodies in the circulatory system of calves prior to the ingestion of colostrum and absorption of antibodies. Many congenital defects in cattle are not recognized until later in life after the window of opportunity for detecting pre-colostral BVDV-specific antibodies is over. This limitation in the identification of TI cattle makes it difficult to correlate congenital defects with TI status. A potential outcome of the methylome data generated in this study would be the selection of biomarkers to identify TI cattle. Identification of TI cattle later in life would allow measurements of growth, health, and productivity to be compared with uninfected herd mates. These analyses would enhance our understanding of the mechanisms by which fetal BVDV infections impact specific organs and metabolic processes.

Pathways in common between TI WBC and PI splenic methylomes include the Human Embryonic Stem Cell Pluripotency pathway which impacts fetal growth and development, and the Role of Macrophages, Fibroblasts, and Endothelial cells in Rheumatoid Arthritis pathway affecting immune cell functions. Similarly, TI and PI methylomes shared some hyper- and hypomethylated genes including *NFATc1* and *WNT7A*, which were hypermethylated in both TI WBC and PI splenic methylomes. However, there are many examples of genes that differ in their methylation status between TI WBCs and PI splenocytes. For example, *CALM1*, which was hypermethylated in TI WBC DNA, was hypomethylated in PI spleen DNA.

### 4.6. Potential Pitfalls, Confounders, and Complications Affecting the Interpretation of Methylome Data

There are several issues that complicate the interpretation of DNA methyl seq data and pathway analyses. First, these data are limited to methylation changes to DNA and do not include other epigenetic alterations such as histone modifications and post-translational regulation by non-coding RNAs. Compounding this issue, DMCs in TI WBC DNA were found in genes impacting histone acetylation, suggesting that histone modifications likely also play a role in gene expression in TI calves. These genes include histone PARylation factor 1 (*HPF1*, LOC101902204) and H1.8 linker histone (*H1-8*), which are hypomethylated in TI WBC DNA compared to controls. Upregulation of *HPF1* would enable chromatin and histone binding, thereby decreasing transcription and promoting DNA damage repair [41]. H1-8, a member of the histone family, interacts with DNA to compact it between nucleosomes inhibiting transcription [42]. Upregulation of these two genes would be expected to have an additive effect causing downregulation of genes through histone acetylation. This action is countered by two genes that contain hypermethylated CpGs in TIs compared to controls, Histone deacetylase 11 (*HDAC11*) and the REST corepressor (*RCOR1*). HDAC11 removes acetyl groups from histones leading to closed chromatin structure and decreased gene expression. Downregulation of *HDAC11* would decrease the deacetylation of histones resulting in a more open chromatin structure potentially allowing increased expression of gene expression [31]. RCOR1 is a component of a complex that functions as a transcriptional repressor by restricting access to the transcriptional machinery [43]. Downregulation of RCOR1 may potentially release its repressor action on transcription. In contrast to the epigenetic effects on *HPF1* and *H1-8*, the downregulation of *HDAC11* and *REST* may enhance the transcription of target genes. Therefore, the effects of TI during postnatal life are complex and difficult to attribute solely to the epigenetic modifications of a single gene.

Several genes contain both hyper- and hypomethylated CpGs. In genes that encode multiple transcripts via alternative splicing, it is possible that hypermethylated CpGs could interfere with the production of one transcript versus another, giving rise to differential expression of “duplicate genes” in the IPA analyses. Examples include *WNT7A*, *WNT7B*, and *PRKCA*, which are predicted to be both upregulated and downregulated based on multiple DMCs. In addition, the methylome data do not specify whether the modified genes are located on the paternal, maternal, or both chromosomes. Inhibition of a gene by methylation may have greater or lesser effects in the developing fetus if the DMG occurs on the paternal chromosome versus the maternal allele [44]. In mice, genes on the maternal chromosomes involved in brain development are expressed during fetal development whereas paternal genes are expressed later [45]. If a hypermethylated gene involved in growth is located on the paternal chromosome, it may be downregulated and unresponsive to stimulation in ~50% of the TI fetuses. The other 50% of TI fetuses may transcribe the hypomethylated allele on the maternal chromosome with transcription and translation of the gene product contributing to normal growth. The added variability in gene expression introduced by parental bias could potentially be perceived as “incomplete penetrance” of an epigenetically modified gene in the phenotype of the offspring. For example, parental bias might explain the range and variation in the TI birth weights.

Hypermethylation and hypomethylation of genes have the potential to inhibit or increase the transcription of the gene during the lifetime of the individual. However, transcription of an individual gene at a given point in time is dependent on the application of a stimulus and the upstream regulatory elements involved in transcription. For example, the hypomethylation of *NFATc2* is predicted to activate *RCAN1* and *NPPB*, which are related to exercise-induced cardiac hypertrophy and concentric hypertrophic cardiomyopathy, respectively. Upregulation of *NFATc2* and activation of *RCAN1* and *NPPB* might only be fulfilled in the presence of mechanical and environmental stressors such as exercise or high altitude, physiological factors such as epinephrine/norepinephrine, and natriuretic peptides in response to excitement and blood pressure. Additional stressors may include increased pulmonary hypertension due to tissue damage caused by microbial infections of the respiratory tract and nutritional deficiencies affecting cardiac muscle function. In the absence of these factors, activation of these three genes might not be realized and the cardiac effects may not develop or be observed.

It is not currently known whether the DMCs and DMRs present in the DNA of WBCs are also differentially methylated in other tissues of TI calves. For example, is a hypermethylated gene in the hepatic fibrosis pathway in the WBC DNA also hypermethylated in DNA from the liver? The occurrence of hypermethylated genes such as the *WNTs* in both the WBC DNA of the TI calves and in the placentae of PI fetuses suggests that some genes may be similarly methylated as a general consequence of fetal BVDV infection regardless of the tissue. This question warrants future investigation of the methylome in other tissues of TI cattle. Finally, the IPA pathway analyses are based on information excerpted from studies of the molecules and pathways in humans, mice, and rats. These genes and pathways have not been as extensively explored in bovine species, perhaps limiting the analyses. To verify the pathways predicted in IPA, the DMCs and DMRs were also analyzed in the bovine genome using KEGG. Overall, similar and identical pathways were identified in KEGG as in IPA analyses, especially with regard to the developing immune system. While it is important to interpret the methylome data in light of these limitations, the methylome data presented here establish a basis for future studies of the anatomic and physiologic changes in the bovine due to fetal TI BVDV infection.

The role of viruses that induce epigenetic changes and impact cellular processes that culminate in serious and chronic diseases has been increasingly the subject of investigation at the cellular and molecular levels [46]. Microbial infections during pregnancy that induce epigenetic changes are of particular interest because of their potential long-term impact on the health of the offspring and perhaps across generations [47]. Mechanisms involved in epigenetic alterations may differ between microorganisms that cross the placenta and those that incite an inflammatory/cytokine response in the dam. In the latter case, cytokines and other proteins may cross the placenta to influence the methylation of fetal DNA [48]. BVDV is part of the first group of pathogens that readily cross the placenta in immunologically naïve dams. The mechanism(s) by which BVDV triggers CpG methylation are likely to be indirect since BVDV replication occurs in the cytoplasm and BVDV-encoded proteins and RNAs do not directly interact with cellular DNA. Alterations in the methylome of PI BVDV fetuses were previously shown to occur in genes associated with the immune system, nervous system, heart, and bone [12]. All of these associations correspond to naturally occurring congenital defects observed in PI calves [12,49]. In this report, extensive epigenetic modifications to the WBC DNA of TI BVDV calves and predictions based on pathway analyses are reported, expanding our knowledge of the range of potential effects of BVDV fetal infections. The actual effects of these DMGs on the function of multiple organs, growth, and metabolism of TI cattle will be determined in future studies.

## 5. Conclusions

In a controlled experiment, late-term fetal BVDV infection resulted in a decrease in the mean birth weight compared to controls and induced an active immune response which cleared the virus. Fetal BVDV infection during late gestation caused epigenomic modifications in the methylome of WBCs collected at birth. Many of these modifications are predicted to impact fetal development, growth, and the immune system by potentially impacting gene expression postnatally with consequences for the health and productivity of TI cattle (Figure 14). Future studies will determine the impact of these epigenomic changes on the growth, immunity, and development of other organs in TI calves.

## Figures and Tables

**Figure 1 viruses-16-00721-f001:**
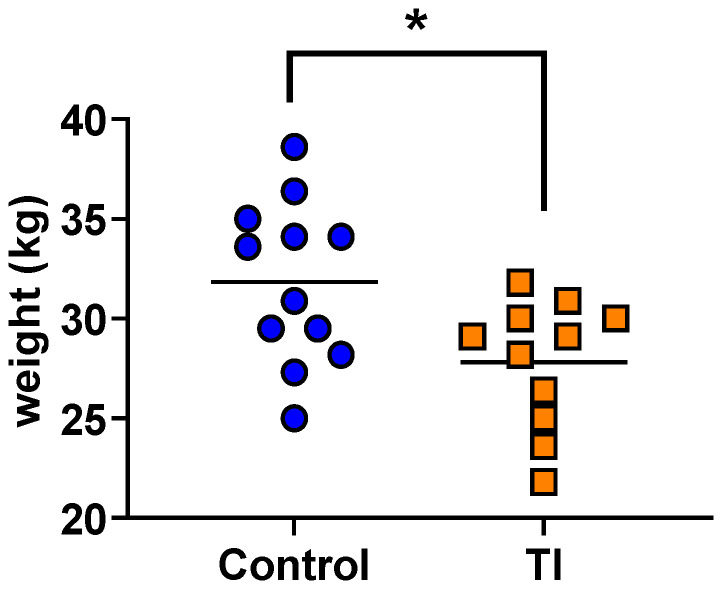
Control and TI heifer calf birth weights with mean (kg), *, *p* < 0.05.

**Figure 2 viruses-16-00721-f002:**
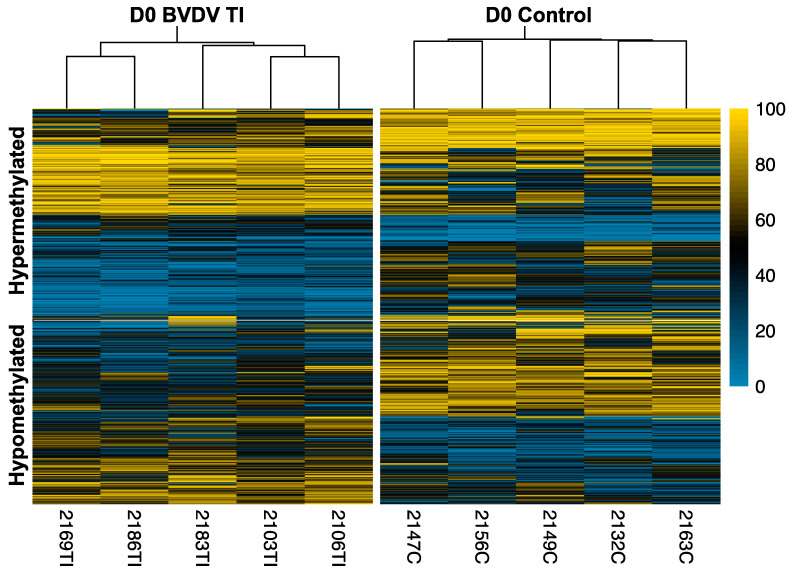
Heatmap of 1072 hypermethylated and 1277 hypomethylated DMCs identified using logistic regression, using a chi-squared test and 25% methylation difference cut-off. Each column represents a replicate for controls and treatment. The horizontal axis shows clustering within the two groups. The yellow color palette indicates higher percent methylation and a density plot on the right shows the distribution of percent methylation values of the map.

**Figure 3 viruses-16-00721-f003:**
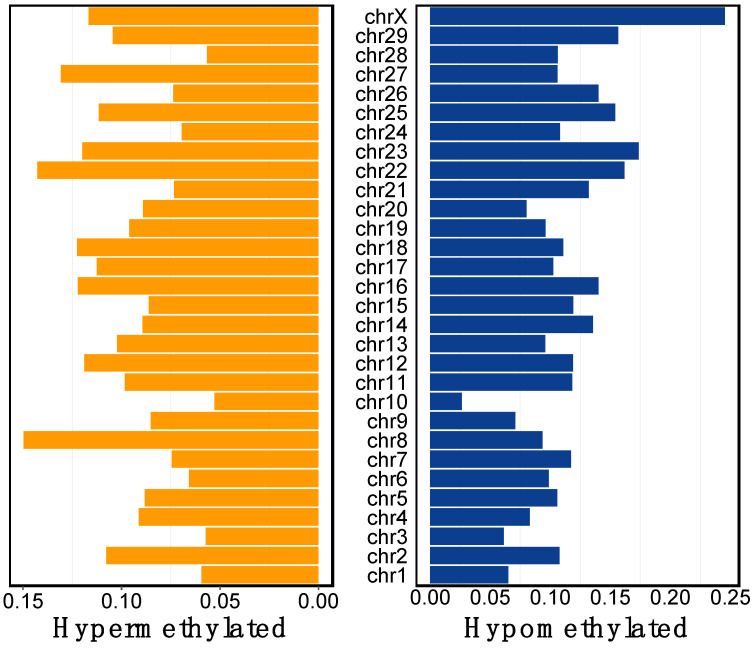
The location and percentage of hyper- and hypomethylated DMCs on each chromosome.

**Figure 4 viruses-16-00721-f004:**
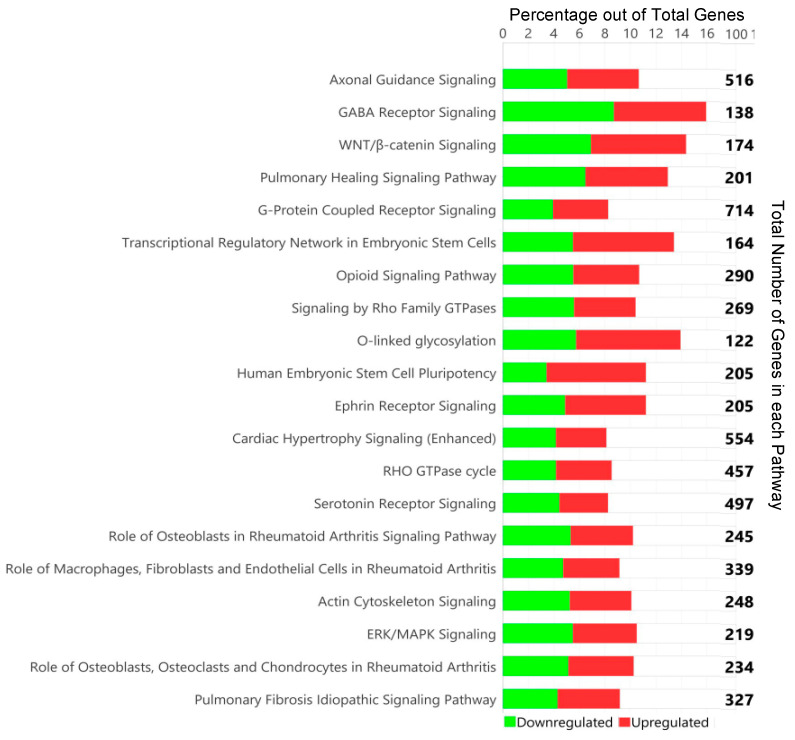
Top twenty canonical pathways in IPA based on DMCs with the total number of DMCs listed. The bars indicate the percentage of hypermethylated genes (shown in green) and hypomethylated genes (shown in red) in the pathway.

**Figure 5 viruses-16-00721-f005:**
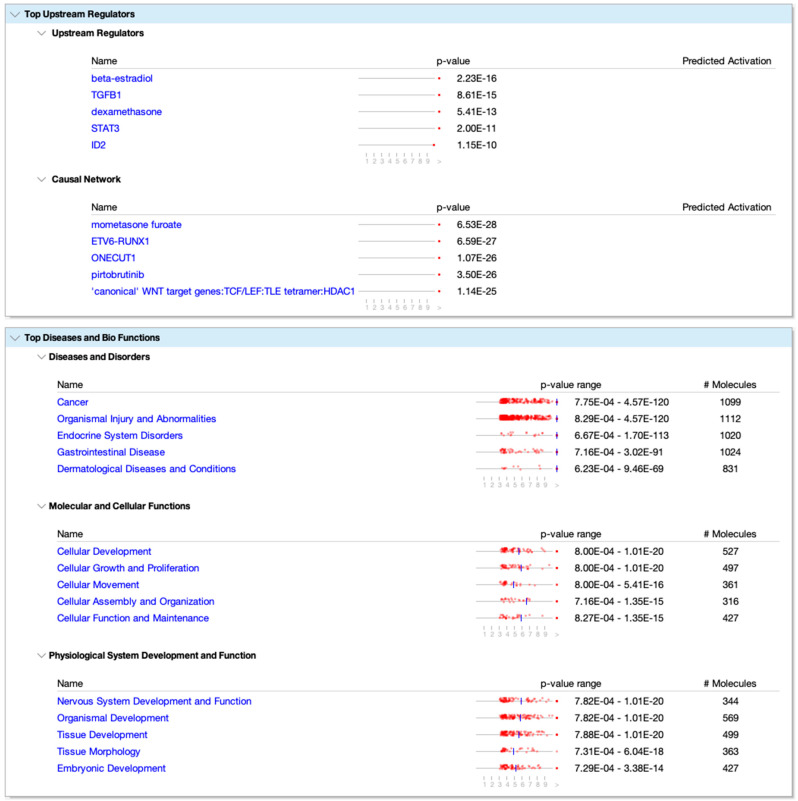
Top upstream regulators and top diseases and functions in IPA based on DMCs. The # Molecules represent the total molecules in the pathway. The *p*-value refers to the upstream regulator or network identified. The *p*-value range is for each disease and biological function pathway listed. Values (1–9) underneath the graphic represent the −log10 (*p*-value). For example, a *p*-value of 1 × 10^−8^ represents a value of 9 [−log10 (*p*-value)] in the scale below the graphic. Symbols represent genes in the pathway with DMCs with *p*-values ranging from 0.001 to 1 × 10^8^. The DMC symbols with *p*-values less than 1 × 10^9^ are not shown in the graphic due to space limitations. The blue vertical line on the scale represents the median of the p-value exponents. If one or more *p*-values are < 10^−10^, then a single dot is shown at the right side of the scale.

**Figure 6 viruses-16-00721-f006:**
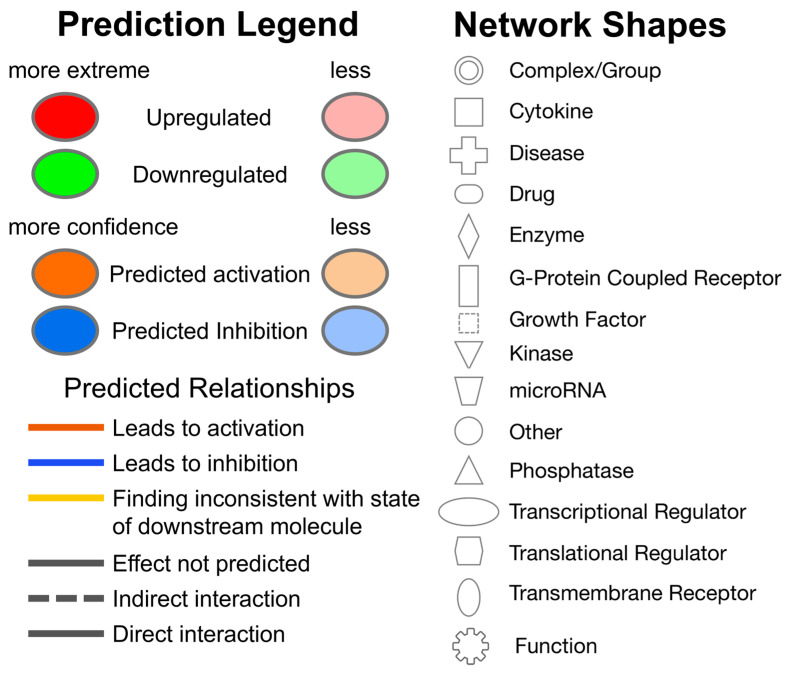
Prediction Legend, Graphical Summary Legend, Network Shapes and Path Designer Shapes for Ingenuity Pathway Analysis. The double pink outlines around a shape indicate the gene is found in the DMC data. See IPA link for more details: https://qiagen.my.salesforce-sites.com/KnowledgeBase/articles/Knowledge/Legend, accessed on 1 March 2024).

**Figure 7 viruses-16-00721-f007:**
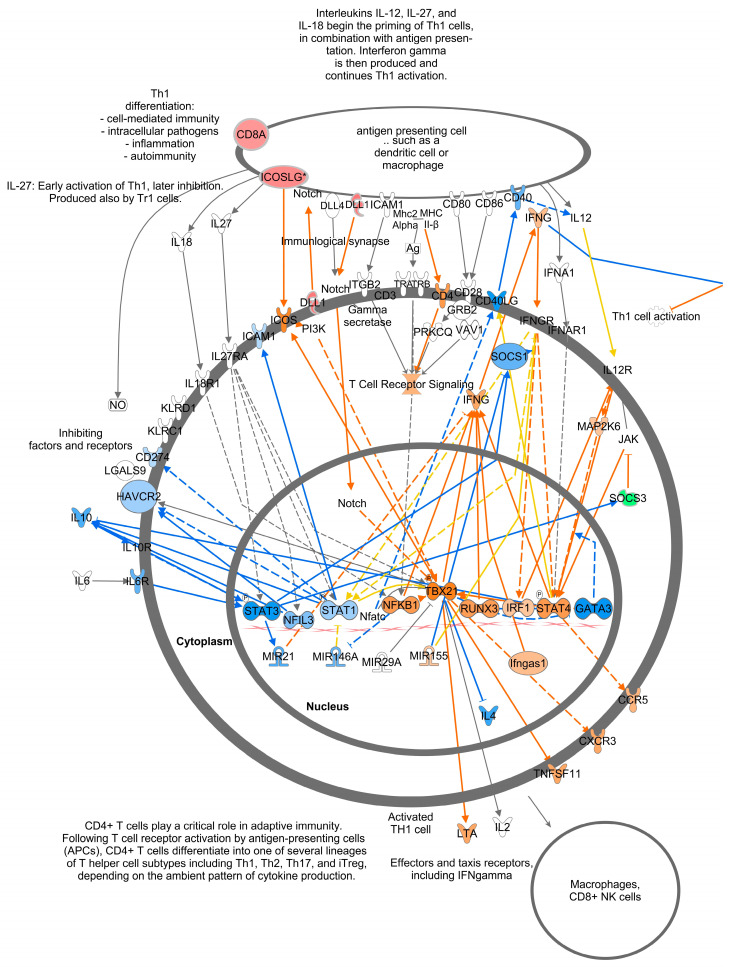
T helper 1 Activation Pathway. There were 173 genes in this pathway with 6 downregulated (or hypermethylated) and 7 upregulated (or hypomethylated) genes. This T helper 1 Pathway overlaps on the upper right-hand side with the T helper cell Pathway described in Figure 8. Genes are predicted to change as indicated in the Prediction Legend (Figure 6).

**Figure 8 viruses-16-00721-f008:**
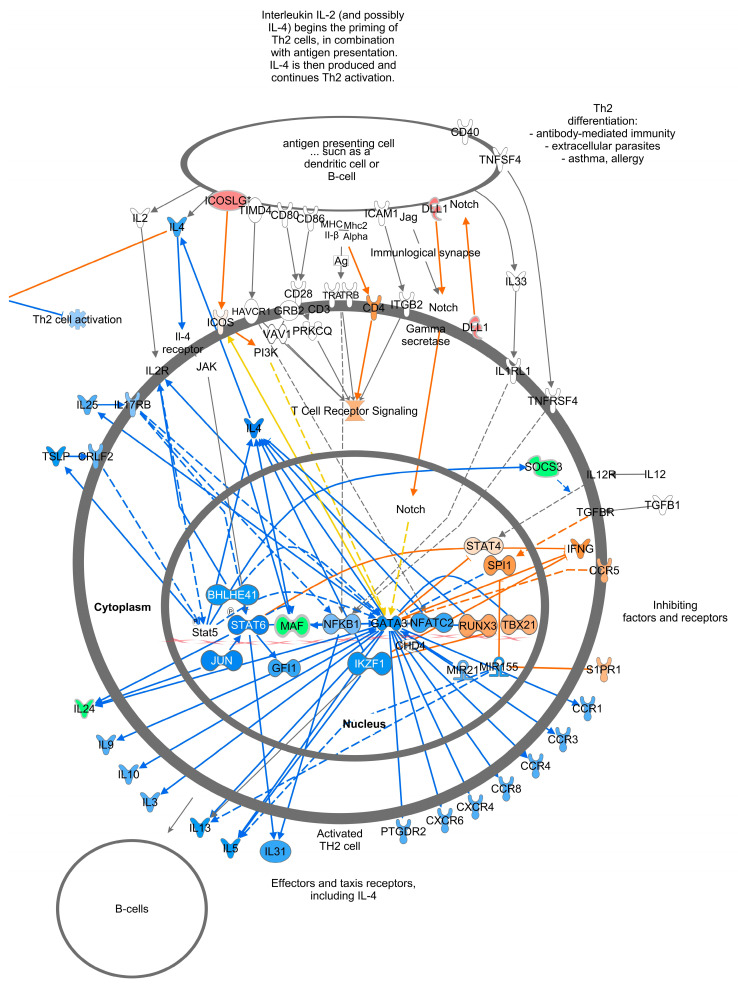
T helper 2 Activation Pathway. Note overlap with Figure 7, in upper left hand side. Genes are predicted to change as indicated in the Prediction Legend (Figure 6).

**Figure 9 viruses-16-00721-f009:**
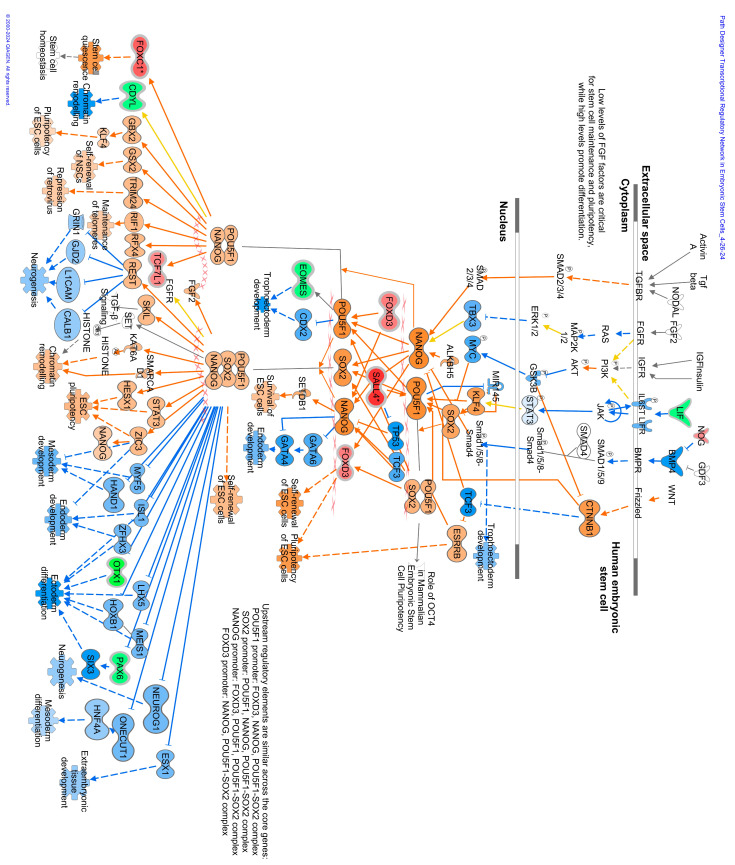
The predicted effects of TI on Transcriptional Regulatory Network in Embryonic Stem Cells in IPA. There were 164 genes in this pathway with 9 downregulated (or hypermethylated) and 13 upregulated (or hypomethylated) genes. Genes are predicted to change as described in the Prediction Legend (Figure 6).

**Figure 10 viruses-16-00721-f010:**
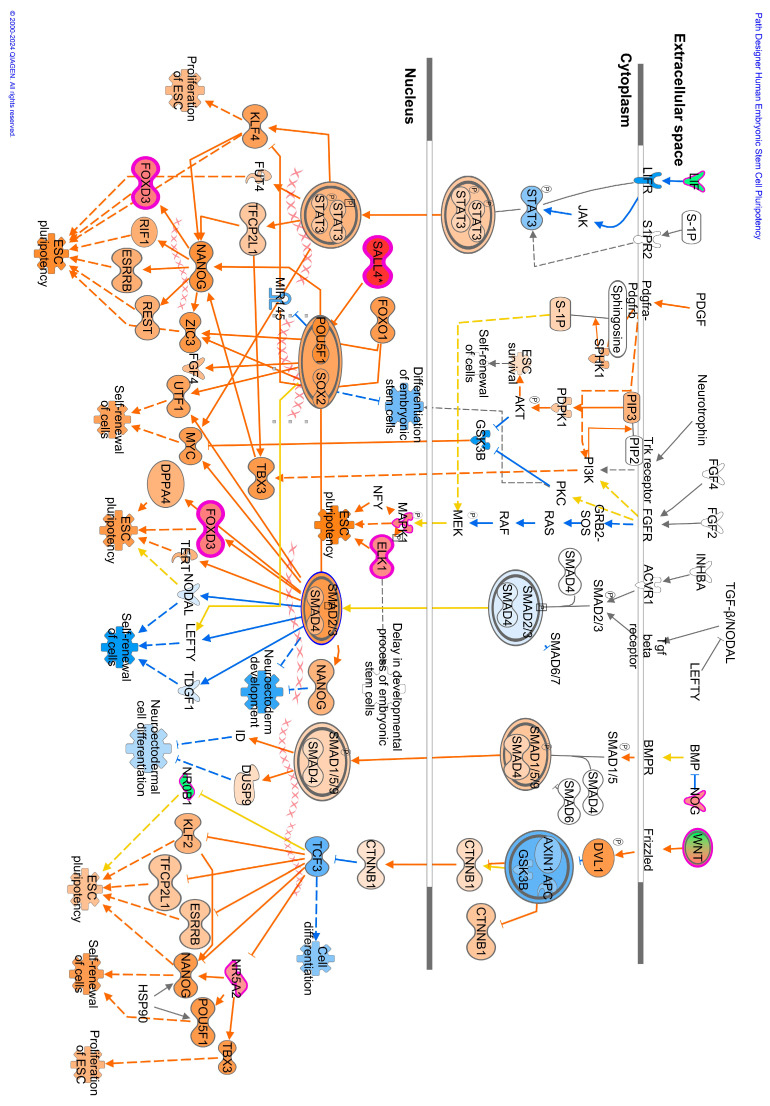
The predicted effects of TI on Human Embryonic Stem Cell Pluripotency in IPA. Genes are predicted to change as described in the Prediction Legend in Figure 6.

**Figure 11 viruses-16-00721-f011:**
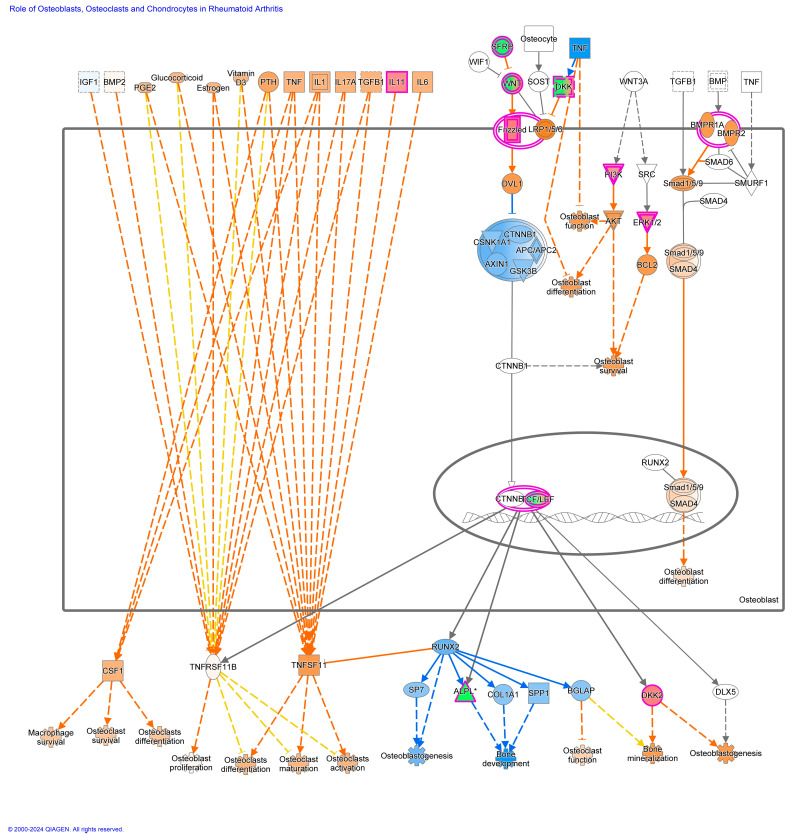
Role of Osteoblasts in Rheumatoid Arthritis Pathway. There were 234 genes in this pathway with 12 downregulated (or hypermethylated) and 12 upregulated (or hypomethylated) genes. Other genes are predicted to change as described in the Prediction Legend in Figure 6. Other predicted upregulated and downregulated genes affecting osteoclasts and chondrocytes in this pathway can be found in Appendix A.

**Figure 12 viruses-16-00721-f012:**
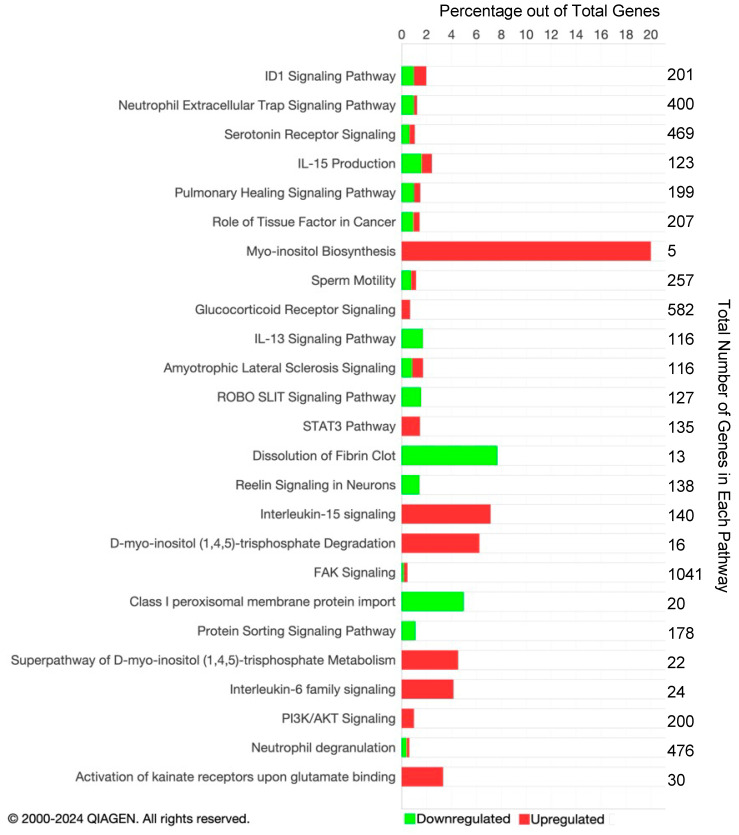
Pathways containing CpGs in promoters predicted in IPA. The bars indicate the percentage of hypermethylated genes (shown in green) and hypomethylated genes (shown in red) in the pathway. The total number of CpGs is listed on the right.

**Figure 13 viruses-16-00721-f013:**
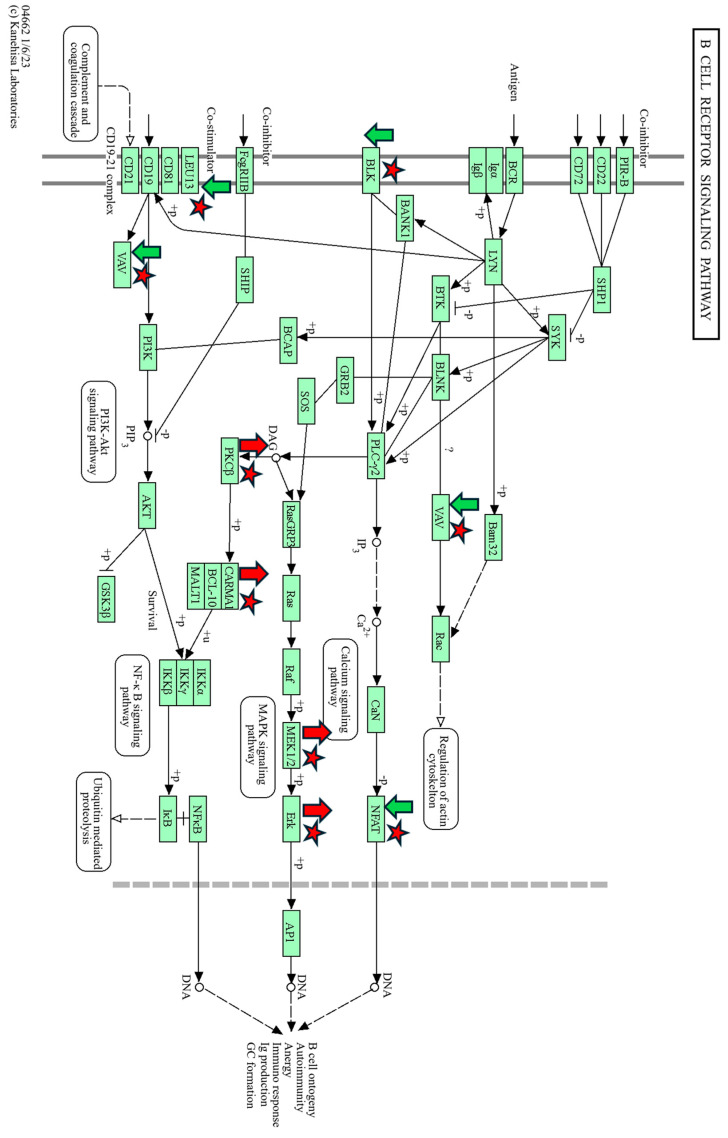
B Cell Receptor Signaling Pathway, KEGG. Differentially methylated genes are marked with red stars, hypermethylated/downregulated genes are indicated by green arrows, and hypomethylated/upregulated genes are indicated by red arrows.

**Figure 14 viruses-16-00721-f014:**
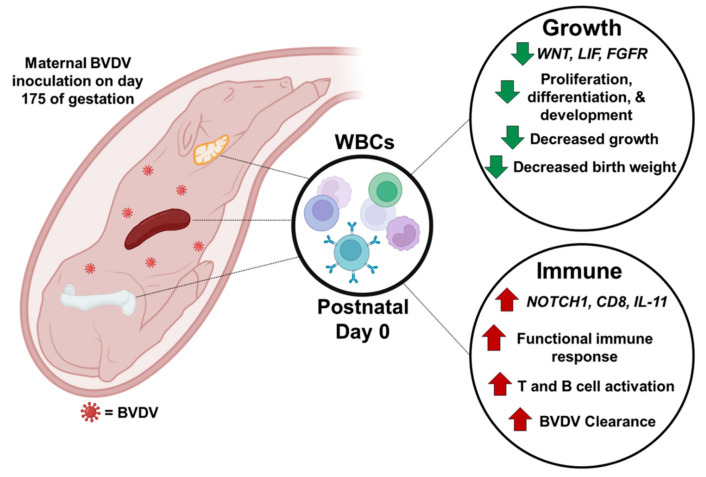
Graphical summary of the epigenetic modifications caused by transient BVDV fetal infection on the WBC methylome obtained at birth. Representative hypermethylated genes such as *WNT*, *LIF*, and *FGFR* are downregulated (indicated by the green arrows), negatively impacting fetal growth and organ development. These epigenetic changes are supported by an observed decrease in the mean weight of TI calves at birth compared to uninfected controls. Hypomethylated genes such as Notch1, CD8, and IL-11 (red arrows) are potentially upregulated due to the fetal immune response to BVDV infection. Activation of these genes with activation of the adaptive immune response is supported by the presence of BVDV-specific antibodies in serum and clearance of the virus by TI calves at birth.

## Data Availability

Raw data such as FASTQ, bedGraph, and processed Excel files are available in the NCBI GEO Database: https://www.ncbi.nlm.nih.gov/geo/browse/ (accession number GSE255721).

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
