# Peer review of "Epigenetic Modifications of White Blood Cell DNA Caused by Transient Fetal Infection with Bovine Viral Diarrhea Virus"

_viruses, 2024, doi:10.3390/v16050721_

Round 1

Reviewer 1 Report

Comments and Suggestions for Authors

Basically my opinion is that this is an OK paper to publish in Viruses.  The main question is if transient BVDV infection causes epigenetic modifications. The study setup is such that the main question is addressed properly and that data are sound. The only shortcoming I see is that epigenetic modifications present in blood cells may not be present in tissues, but the authors realize that. The dataset is very extensive and the paper adds new knowledge to the field of pestivirus infection. The conclusions are sound. There is really a lot of information in the paper and I think the authors have done a great job. It is inevitable that some figures with signaling pathways are difficult to read, Figs 5-9, and there is speculation. However, the speculation on possible signaling pathways affected is the basis for new experimental studies that can further elucidate the many ways that viruses use to affect their hosts (in their benefit).  The paper reads really well and I enjoyed it. Did not observe a single type but might have missed it.

Reviewer 2 Report

Comments and Suggestions for Authors

The manuscript by Campen et al. compared epigenetic modifications of white blood cell DNA by methylome in calves infected with BVDV. The authors have tried to understand how BVDV infection regulate the epigenetic modifications of white blood cell DNA, gene expression, signaling pathways and immune functions. However, results of this manuscript were all predicted from the methylome data without any experimental validation to support the mechanism of BVDV infection regulate gene expression and signaling pathways. This is a major drawback of this manuscript.

Specific comments:

1.     Suggest testing several DMRs located in promoters in results 3.3. to further support methylome data.

2.     Suggest validating genes expression in result 3.4. and 3.5. which involved in immune system activation pathways (overlapped pathways analyzed by IPA and KEGG).

3.     Line 263: 3.3. should be 3.2.

4.     Line 300: 3.4. should be 3.3.

5.     Line 502: 3.5. shoud be 3.4.

6.     Figure 10, the total CpGs numbers were not fully displayed.

Reviewer 3 Report

Comments and Suggestions for Authors

Overall the manuscript is well written, the methods are clearly described, the results are well presented and the conclusions drawn are supported by the data.

A minor point is the placement of the figure legends to the left of the panels. This is quite distracting in some instances where the main text overlaps. Suggest modifying to the more conventional format of legends below the relevant panels. This would also enable the figure panels to be larger.

Line 41 Suggest revision “Fetal BVDV infection between day 30 and 125 of gestation”

Although the “window” for PI development is difficult to precisely define, I would suggest that in the first 30 days, embryonic death is the most likely outcome, PIs between day 30 and 125, and then abortion/deformities/normal for the remainder. All the while accepting that numerous factors could affect these “windows”.

Line 139-140 The sentence seems incomplete “Blood was collected in XXX and”. It is also not clear why these details differ from those described in Lines 134 to 135 for serum harvesting. Consider deleting the first sentence or revise as appropriate.

 Line 254 Consider providing the RT-PCR results as a supplemental figure.

 Line 272 Figure 2 – if possible, the label on the left panel should be changed to “D0 BVDV TI”.

 Line 316 Table 1 – the aspects of this table seem to be more like figure panels. Consider converting to a figure.

In any case there are some aspects of each panel that are not clearly explained and they should be either in the table title or the figure legend (if it is changed to that).

There are line elements in each with red dots in each. The lines seem to have scales/numbers associated with them however they are indistinct. Can this be improved? I presume that they are associated with the final “column” on the right. In any case, all elements should be clearly explained.

 Line 332 Figure 4 – Is the label on the top of the figure required? Two elements are not shown in the figure nor mentioned in the legend.

Figure 363 Some elements are not explained in the “Prediction legend” – for example dotted lines, purple outlines, and white shapes. Is there any importance to gene “shapes”? I.e squares, ovals, triangles etc. Any specific meaning should be explained.

Round 2

Reviewer 2 Report

Comments and Suggestions for Authors

The manuscript has been sufficiently improved to warrant publication in Viruses.

Author Response

There are no notes needing response from authors regarding necessary revisions. Thank you for accepting the manuscript.